# CAVPENET Peptide Inhibits Prostate Cancer Cells Proliferation and Migration through PP1γ-Dependent Inhibition of AKT Signaling

**DOI:** 10.3390/pharmaceutics16091199

**Published:** 2024-09-12

**Authors:** Bárbara Matos, Antoniel A. S. Gomes, Raquel Bernardino, Marco G. Alves, John Howl, Carmen Jerónimo, Margarida Fardilha

**Affiliations:** 1Laboratory of Signal Transduction, Department of Medical Sciences, iBiMED-Institute of Biomedicine, University of Aveiro, 3810-193 Aveiro, Portugal; barbaracostamatos@ua.pt (B.M.); mfardilha@ua.pt (M.F.); 2Cancer Biology and Epigenetics Group, IPO Porto Research Center (CI-IPOP), RISE@CI-IPOP (Health Research Network), Portuguese Oncology Institute of Porto (IPO Porto), Porto Comprehensive Cancer Center Raquel Seruca (Porto.CCC), R. Dr. António Bernardino de Almeida, 4200-072 Porto, Portugal; carmenjeronimo@ipoporto.min-saude.pt; 3Department of Biophysics & Pharmacology, Institute of Biosciences of Botucatu, São Paulo State University, Botucatu 18610-034, SP, Brazil; antoniel.gomes@unesp.br; 4Unit for Multidisciplinary Research in Biomedicine (UMIB), School of Medicine and Biomedical Sciences (ICBAS), University of Porto, Rua Jorge de Viterbo Ferreira 228, 4050-313 Porto, Portugal; raquellbernardino@gmail.com; 5Laboratory for Integrative and Translational Research in Population Health (ITR), 4050-600 Porto, Portugal; 6Department of Medical Sciences, iBiMED-Institute of Biomedicine, University of Aveiro, 3810-193 Aveiro, Portugal; marcoalves@ua.pt; 7Faculty of Health, Education and Life Sciences, Birmingham City University, Edgbaston, Birmingham B15 3TN, UK; 8Department of Pathology and Molecular Immunology, ICBAS-School of Medicine and Biomedical Sciences, University of Porto, Rua de Jorge Viterbo Ferreira, 228, 4050-313 Porto, Portugal

**Keywords:** prostate cancer, treatment, protein-protein interaction, protein phosphatase 1, PP1-targeting peptide

## Abstract

Protein phosphatase 1 (PP1) complexes have emerged as promising targets for anticancer therapies. The ability of peptides to mimic PP1-docking motifs, and so modulate interactions with regulatory factors, has enabled the creation of highly selective modulators of PP1-dependent cellular processes that promote tumor growth. The major objective of this study was to develop a novel bioactive cell-penetrating peptide (bioportide), which, by mimicking the PP1-binding motif of caveolin-1 (CAV1), would regulate PP1 activity, to hinder prostate cancer (PCa) progression. The designed bioportide, herein designated CAVPENET, and a scrambled homologue, were synthesized using microwave-assisted solid-phase methodologies and evaluated using PCa cell lines. Our findings indicate that CAVPENET successfully entered PCa cells to influence both viability and migration. This tumor suppressor activity of CAVPENET was attributed to inhibition of AKT signaling, a consequence of increased PP1γ activity. This led to the suppression of glycolytic metabolism and alteration in lipid metabolism, collectively representing the primary mechanism responsible for the anticancer properties of CAVPENET. Our results underscore the potential of the designed peptide as a novel therapy for PCa patients, setting the stage for further testing in more advanced models to fully realize its therapeutic promise.

## 1. Introduction

Prostate cancer (PCa) is the second most common type of cancer diagnosed in men worldwide, with a total of 1.47 million new cases reported in 2022 [1]. Despite several therapeutic options available for these patients, the development of castration-resistant prostate cancer (CRCP) in a non-negligible number of cases continues to challenge the scientific and medical communities, requiring the development of novel therapeutic approaches.

The important role of serine/threonine-protein phosphatase 1 (PP1) in several tumor-associated biological processes identified it as a potential drug target in cancer [2]. PP1 is a major protein phosphatase, responsible for most dephosphorylation reactions in human cells and critical roles in a wide range of cellular processes [3]. Its catalytic subunit is encoded by the genes *PPP1CA*, *PPP1CB* and *PPP1CC*, producing the respective ubiquitously expressed protein isoforms PP1α, PP1β and PP1γ [4]. Both the activity and intracellular distribution of PP1 are determined by the interaction with regulatory proteins—regulatory interactors of protein phosphatase 1 (RIPPOs). Hence, whilst the direct inhibition of the PP1 active site may produce controversial results [5], targeting specific PP1 holoenzymes seems to be a promising and more selective strategy. However, the characterization of PP1 complexes in cancer is relatively limited, and to date, only 38 PP1 complexes have been functionally characterized in cancer models, most of which are not fully understood (reviewed by Matos et al. 2021 [6]).

Caveolin-1 (CAV1) has been associated with PCa growth and its tumor promoter roles seem to be mediated by the interaction and consequent inhibition of PP1 activity. The CAV1-associated inhibition of PP1 increases the phosphorylation of various substrates, including ERK1/2, PDK1 and protein kinase B (AKT), which contributes to the survival, proliferation and invasive ability of cancer cells [7]. The CAV1-induced metabolic reprogramming of cancer cells, by increasing their glycolytic rate, was also associated with migration and invasion [8].

In recent years, small-molecule drugs have been developed to prevent PP1 complexes; these include both salubrinal [9] and trichostatin A [10]. More recently, a few studies have suggested peptides as a more selective and improved strategy [11]. However, only a few studies have elucidated the use of peptides, based on the RVxF motif, to target and disrupt interactions between PP1 and RIPPOs, both in in vitro [12,13,14] and in vivo scenarios. The latter involves primary mouse tissue [15,16] and heart failure patient tissue samples [17]. In the realm of cancer research, promising in vitro [18,19] and in vivo [20] investigations employed peptides to interrupt the PP1/GADD34 interaction.

Peptide therapeutics have gained significant recognition in recent years. A notable example in the treatment of PCa is the peptide-based therapeutic agent ^177^Lu-vipivotide terraxetan, recently approved by the FDA. This anticancer drug selectively targets the prostate-specific membrane antigen (PSMA) and delivers beta-radiations to effectively destroy prostate cancer cells, underscoring the potential of peptide-based therapies in oncology [21]. Moreover, cell-penetrating peptides (CPPs) have emerged as a viable class of pharmacokinetic modifiers to promote the intracellular delivery of bioactive agents [22]. Moreover, bioactive CPPs (named bioportides) are a powerful technology for the development of therapeutic agents and novel biologics. Thus, bioportides combine the inherent ability of CPPs to cross the plasma membrane with the capacity to modulate intracellular functions, often through a dominant-negative influence upon protein–protein interactions [23]. The application of bioactive CPPs in cancer has taken the first steps and was reviewed by Jauset et al., 2019 [24]. The valuable impact of CPPs in the therapeutic field was recently highlighted by the first FDA-approved CPP-delivered drug—DAXXIFY^TM^ (https://www.fda.gov/drugs/new-drugs-fda-cders-new-molecular-entities-and-new-therapeutic-biological-products/novel-drug-approvals-2022, accessed on 27 October 2023). Moreover, Guergnon et al., 2006 opened up a new direction for CPP-based therapies by introducing the use of CPPs with binding sequences for targeting phosphatases: PP1 or PP2A [25].

In the present study, a peptide based on the PP1-docking motif of CAV1, coupled to the CPP vector penetratin [26], was designed and synthesized to form a sychnologic bioportide—CAVPENET. This study compared the ability of the CAVPENET bioportide and a scrambled homolog (CAVPENET control) to translocate into PCa cells and impact on PCa biology. Related investigations assessed the modulation of PP1 complex-associated signaling pathways and evaluated the structure and dynamics of bioportides bound to PP1. This work further elucidates the potential use of bioportides to treat PCa, by targeting PP1/RIPPO(s) interaction(s).

## 2. Materials and Methods

### 2.1. Bioportides Design and Synthesis

The CAV1 sequence was searched for PP1-docking motifs using the ScanProsite online tool (available at https://prosite.expasy.org/scanprosite/, accessed on 29 November 2019). In brief, the CAV1 sequence was retrieved from Uniprot (Uniprot ID: Q03135) and inputted into ScanProsite. The tool scanned for all types of PP1-binding motifs, identifying the CAV1 PP1-binding motif as an RVxF motif, characterized by the pattern: [RK]-X(0,1)-[VI]-{P}-[FW]. The peptide sequence used for the construction of the bioportide CAVPENET (Table 1) was based on the identified PP1-binding motif of CAV1 (^65^KIDF^68^). The flanking sequences to this motif (^63^VV^64^ and ^69^ED^70^), were also included to provide specificity to the PP1/CAV1 interaction. The resulting protein mimetic motif was covalently coupled to the Penetratin sequence (RQIKIWFQNRRMKWKK), a CPP that is readily internalized into cancer cells [27]. A control bioportide, with a scrambled PP1-binding motif (DFIK) designed to decrease binding specificity, was also designed (Table 1).

The bioportides with the sequences described in Table 1 were prepared by Microwave-assisted solid phase peptide synthesis, using the Liberty Blue^TM^ Automated Microwave Peptide Synthesizer (CEM Microwave Technology, Buckingham, UK), as described in [28,29]. In brief, 0.1 mmol of rink-amide methylbenzylhydrylamine resin was used as the solid phase to support peptide synthesis. After deprotection of the resin using 20% (*v*/*v*) piperidine in N,N-dimethylformamide (DMF), N-α-Fmoc-protected aminoacids (0.2 M), prepared in DMF, were sequentially coupled to the sequence. One molar equivalent of N,N-diisopropylcarbodiimide/OxymaPure^®^ was used as a condensation reagent. The addition of 0.1 molar equivalent N,N-diisopropylethylamine (DIPEA) to coupling reactions reduced racemization and increased synthetic yields. A double coupling strategy was employed to introduce Arg (30 W/75 °C/300 s) without δ-lactam formation. Between each coupling reaction, a deprotection step with 20% piperidine removed Fmoc protecting groups. For cleavage, peptides were incubated for 3 h at room temperature with gentle agitation with a standard cleavage solution: trifluoroacetic acid/triisopropylsilane/water (95:2.5:2.5) Fluorescent analogs of the bioportides to be used in cell imaging uptake assays were also synthesized by aminoterminal acylation with 6-carboxy-tetramethylrhodamine (TAMRA). All bioportides were purified to >95% by semi-preparative scale high-performance liquid chromatography (HPLC) on a PerkinElmer^®^ FlexarTM with UV/Vis detector system and a C18 column (Zorbax 300SB-C18). The purified bioportides were freeze dried and stored at −20 °C until use. The predicted masses were confirmed by mass spectrometry with an accuracy of plus/minus 1 Da.

### 2.2. Cell Culture and Treatments

Human prostate cancer cells (PC-3 (androgen-independent) and LnCaP (androgen-dependent) cell lines) were grown in Roswell Park Memorial Institute media-1640 with L-glutamine (RPMI-1640, Gibco, Life Technologies, Waltham, MA, USA), supplemented with 10% (*v*/*v*) fetal bovine serum (FBS, Gibco, Life Technologies, USA) and 1% (*v*/*v*) penicillin/streptomycin mixture (Gibco, Life Technologies, USA). The cells (passage number 10–16) were maintained in an incubator (MCO-170AICUV, Panasonic, Langen, Germany) at 37 °C, with a 5% CO_2_ atmosphere under fully humidified conditions. The CAVPENET control and CAVPENET bioportides were dissolved in distilled water as concentrated stocks of 100 μM. The stocks were diluted to the desired concentration with RPMI-1640 medium when needed.

Cantharidin (ALX-270-063-M025) was dissolved in dimethyl sulfoxide (DMSO) to 100 μM stocks and diluted in RPMI-1640 medium to the desired final concentrations.

### 2.3. Microscopic Evaluation of Cellular Uptake

To evaluate the intracellular accumulation of bioportides, PC-3 and LnCaP cells were seeded in a 24-well plate (6.00 × 10^4^ and 7.50 × 10^4^ cells/well, respectively) and incubated for 24 h at 37 °C under a 5% CO_2_ atmosphere. After washing cells with phosphate buffered saline (PBS-1×), the growth medium was exchanged for phenol red-free RPMI-1640 medium containing 5µM of TAMRA-labeled CAVPENET or CAVPENET control and incubated for 1 h. After the incubation time, the cells were washed in phenol red-free RPMI-1640 medium and observed by EVOS^TM^ M5000 Imaging, and photos were obtained. Live-cell imaging was used to avoid fixation artifacts. Untreated cells were included as background reading.

### 2.4. Cell Viability Assay

The effect of the CAVPENET bioportides on cell viability was assessed using the PrestoBlue viability assay (Invitrogen, Life Technologies, Waltham, MA, USA). PC-3 and LnCaP cells were seeded in 96-well plates (1.00 × 10^4^ and 1.50 × 10^4^ cells/well, respectively) and incubated at 37 °C, 5% CO_2_ atmosphere for 24 h. Then, the growth medium was exchanged for fresh medium without FBS and containing different concentrations of each bioportide (5, 10 and 20 µM of CAVPENET and CAVPENET control) and incubated for 48 h. Untreated cells were also included in the assay and used as the control (100% viability). One hour before the incubation time, 10 µL of PrestoBlue^TM^ Cell Viability reagent, resazurin-based reagent (Thermo Fisher Scientific, Waltham, MA, USA) was added to each well and incubated for 1 h at 37 °C and 5% CO_2_ atmosphere. After the incubation, 100 µL of the culture medium of each well was transferred to a black bottom 96-well plate and the fluorescence at λ_abs_ = 560 nm and λ_em_ = 590 nm was measured in the microplate reader Tecan Infinite^®^ 200 PROseries, Mannedorf, Switzerland. The fluorescent signal results from the reduction of resazurin to resorufin. Cell viability was measured as the percentage of viable cells, considering untreated cells as 100% viability. Four independent experiments with five replicates for each condition were performed.

### 2.5. Wound Healing Assay

The influence of bioportides upon PCa cell migration was determined using the wound healing assay. In brief, PC-3 and LnCaP cells were seeded in a 24-well plate (1.00 × 10^5^ and 1.20 × 10^5^ cells/well, respectively) and incubated at 37 °C, 5% CO_2_ atmosphere, for 24 h. Confluent cell monolayers were wounded by scratching lines with a 200µL pipette tip. The medium was removed, and the cells were washed with PBS1x and incubated with fresh medium without FBS containing 10 µM of CAVPENET control or CAVPENET for 48 h (PC-3 cells) or 72 h (LnCaP cells). Untreated cells were also included in the assay. Photographs were taken under ×40 magnification, using the EVOS^TM^ M5000 Imaging immediately after wound incision and after 48 h or 72 h (depending on the cell line). The results were expressed as a % of wound closure, relative to the time 0 h. Three independent replicates were performed for each condition.

### 2.6. Immunoblotting

PC-3 and LnCaP cells in six-well plates (2.5 × 10^5^ and 3.5 × 10^5^ cells per well, respectively) were treated with various concentrations of CAVPENET and CAVPENET control (10 and 20 µM) in FBS-free RPMI-1640 culture medium at 37 °C under a 5% CO_2_ atmosphere for 48 h. After experimental treatment, cells were collected and lysed using 1× radioimmunoprecipitation assay lysis buffer (RIPA: 0.5 M Tris-HCl, pH 7.4, 1.5 M NaCl, 2.5% deoxycholic acid, 10% NP-40, 10 mM EDTA; Millipore, Temeula, CA, USA), supplemented with 1× protease and phosphatase inhibitor cocktails (Thermo Scientific, Waltham, MA, USA). The lysed cells were centrifuged at 12,000× *g* for 10 min at 4 °C and the supernatants were collected (cell lysates). The protein concentration of cell lysates was determined using the bicinchoninic acid assay (BCA; Pierce Thermo Fisher Scientific, Inc., USA) with bovine serum albumin (BSA) as standard. Equivalent amounts of protein from each lysate (25 µg) were resolved by 12% or 15% sodium dodecyl sulfate-polyacrylamide gel electrophoresis (SDS-PAGE) and then transferred onto nitrocellulose membranes (Amersham^TM^ Protan^®^ NC 0.45 µm, GE Healthcare, Chicago, IL, USA). The efficiency of the transference was confirmed using PonceauS staining, which was also used as the loading control. After blocking the non-specific binding-sites with 5% non-fat milk or 5% BSA in 1× Tris-buffered saline (TBS: 10 nM Tris-HCl, 150 mM NaCl, pH 8.0) containing 0.1% (*v*/*v*) Tween-20 (TBS-T) for 1 h at RT, the membranes were incubated with primary antibodies. The primary antibodies were diluted in either 5%non-fat milk or 5% BSA and incubated for 1 h or ON at 4 °C: anti-p(Ser473)-AKT (rabbit, 1:2000, 4060S, Cell Signaling), anti-AKT (mouse, 1:500, sc-5298, Santa Cruz Technologies, Santa Cruz, CA, USA), anti-p(Ser9)-GSK3β (mouse, 1:1000, sc-373800, Santa Cruz Biotechnologies, Santa Cruz, CA, USA), anti-GSK3β (rabbit, 1:1000, 9315S, Cell Signaling), anti-p(Thr320)-PP1α (rabbit, 1:1000, ab62334, Abcam, Cambridge, UK), anti-PP1α (rabbit, 1:2500, homemade), anti-PP1β (mouse, 1:1000, sc-365678, Santa Cruz Biotechnologies), anti-PP1γ (rabbit, 1:1000, homemade), anti-p(Thr37/46)-4E-BP1 (rabbit, 1:1000, 236B4, Cell Signaling), anti-4E-BP1 (rabbit, 1:1000), anti-HK2 (mouse, 1:500, sc-374091, Santa Cruz Biotechnologies), anti-FASN (mouse, 1:500, sc-55580, Santa Cruz Biotechnologies); anti-CD36 (rabbit, 1:2000, NB400-144, Novus Biologicals, Centennial, CO, USA) and anti-CPT1A (mouse, 1:500, sc-393070, Santa Cruz Biotechnologies). Then, the membranes were washed three times with TBS-T, 10 min and incubated with the respective secondary antibodies IRDye^®^68RD anti-rabbit (926-68071), 1:10,000 and IRDye^®^800CW anti-mouse (926-32210), 1:10,000, LI-COR Biosciences (Licon, NE, USA) for 1 h at RT. After washing two times with TBS-T and one time with TBS, the membranes were revealed using the Odyssey Infrared Imaging System (LI-COR^®^ Biosciences, Lincoln, NE, USA). The quantitative analysis was carried out using the ImageLab^TM^ software, version 6.1 (Biorad, Hercules, CA, USA). The results were normalized to PonceauS staining and the fold change relative to the untreated cells was represented.

### 2.7. Molecular Modeling

To study the interaction between PP1 and CAVPENET bioportides at the atomic level, their complexes were predicted using AlphaFold Multimer [30]. The best model for each case was selected according to the highest Local Distance Different Test (lDDT) value indicated by AlphaFold. To prevent high spatial fluctuations of unfolded terminations, the range of residues Leu7-Pro298 of PP1 was considered. Each complex was submitted to Molecular Dynamics (MDs) simulations using GROMACS v.2022.2 [31] under the Charmm36 m force field [32]. The protonation state of PP1 at a physiological pH of 7.4 was predicted by the PROPKA3 web server [33]. Magnesium ion (Mg^2+^) coordinates were obtained by superposing the X-ray structure (PDB id 4G9J) onto the predicted models.

The CHARMM-GUI web server was used to build all systems [34]. The C-terminal of each peptide was capped with an additional neutral alanine to prevent any undesired fluctuation or electrostatic effects of the last residue within the vicinity of the protein. Each system was solvated with TIP3 water models and neutralized with NaCl to reach a concentration of 0.15 M. Each system was minimized with the Steepest Descent algorithm until reaching an energy below 100 kJ/mol/nm^2^, and then five equilibration steps were performed with an NVT ensemble of 500 ps at a time step of 0.001 picoseconds (ps), controlling the temperature of the system at 310.15 K with the V-Rescale thermostat [35] with a time constant of 0.1 ps, and then, an NPT step of 1 ns was performed at a time step of 0.002 ps and thermostat time constant of 1 ps, monitoring the pressure of the system at 1 bar using the Parrinello–Rahman barostat [36] with a time constant of 1 ps. In both steps, backbone and sidechain heavy atoms were position restrained at a respective force of 400 and 40 kJ/mol/nm^2^. In the following step, the barostat was monitoring the pressure with a time constant of 5 ps, and position restraints were reduced to 200 and 20 kJ/mol/nm^2^ for backbone and side chain heavy atoms, respectively, during 5 ns. Two additional steps of 5 ns were performed, decreasing position restraints to 100 and 10, and 50 and 5 kJ/mol/nm^2^ for backbone and side chain heavy atoms, respectively. The production was performed in three independent replicas per system for 200 ns, collecting frames every 20 ps.

Non-bonded interactions were calculated up to 10 Å with a switching-force function between 10 and 12 Å. Long-range electrostatics interactions were calculated within a 12 Å cut-off, using the particle-mesh Ewald (PME) method. Root-mean-square deviation (RMSD) and root-mean-square fluctuation (RMSF) analyses were carried out using GROMACS built-in functions, and atomic distances and the percentage of contacts were calculated using VMD scripting [37].

### 2.8. Small Interfering RNA Transfection

To seek the involvement of PP1 in CAVPENET-induced reduction in the PCa cells’ viability, knockdowns of specific PP1 isoforms were employed and combined with peptide treatment. Briefly, PC-3 cells were plated in a 96-well plate (10,000 cells/well) and transfected with 15 pmol small interfering RNAs (siRNAs): siRNA *PPP1CA* (Dharmacon, Lafayette, CO, USA, Sequence: CCGCAUCUAUGGUUUCUACdTdT), siRNA *PPP1CB* (Dharmacon, Sequence: UUAUGAGACCUACUGAUGUdTdT) or siRNA *PPP1CC* (Dharmacon, Sequence: GCAUGAUUUGGAUCUUAUAdTdT). Concomitantly, non-targeting siRNA was transfected as a negative control. The Lipofectamine RNAiMAX reagent was used as the transfection reagent, following the manufacturers’ instructions. After 24 h, the cells were treated with 10 μM of CAVPENET for 48 h at 37 °C and 5% CO_2_ atmosphere. Untreated cells were included for each siRNA condition. Cellular viability was evaluated using the PrestoBlue^TM^ Cell Viability reagent, as described above (topic 2.4).

### 2.9. Seahorse Assay

The Seahorse metabolic assay was used to evaluate the impact of CAVPENET on the two main metabolic pathways: mitochondrial respiration, through the determination of oxygen consumption rate (OCR); and glycolysis, by measuring the extracellular acidification rate (ECAR). For this purpose, PC-3 cells were seeded (25,000 cells per well) on Agilent Seahorse XF24 cell culture microplates in RPMI-1640 with L-glutamine, supplemented with 10% FBS and 1% (*v*/*v*) penicillin/streptomycin (Gibco, Life Technologies, USA) at 37 °C and a 5% CO_2_ atmosphere. After 24 h, the growth medium was exchanged for fresh medium without FBS and containing 10 µM of CAVPENET bioportides and then incubated for 48 h. Cells were then assayed using the Agilent Seahorse XF Cell Mito Stress Test Kit (Agilent Technologies, Santa Clara, CA, USA) according to manufacturer instructions. The OCR was determined after successive injections of oligomycin (inhibits ATP synthase (complex V)), carbonyl cyanide-4 (trifluoromethoxy) phenylhydrazone (FCCP) (uncoupling agent, collapses the proton gradient) and rotenone/antimycin A (inhibits mitochondrial complex I and III). The ECAR was also calculated during the time. Data were normalized to the total protein amount per well. To determine the protein concentration, the cells were lysed using 1x radioimmunoprecipitation assay lysis buffer (RIPA: 0.5 M Tris-HCl, pH 7.4, 1.5 M NaCl, 2.5% deoxycholic acid, 10% NP-40, 10 mM EDTA; Millipore, Temeula, CA, USA), supplemented with 1x protease inhibitors cocktail (Thermo Scientific, USA). The protein concentration of cell lysates was determined using the bicinchoninic acid assay (BCA; Pierce Thermo Fisher Scientific, Inc., USA) with bovine serum albumin (BSA) as standard. Several mitochondrial functional parameters, including basal respiration, ATP production, H^+^ (proton) leak, maximal respiration, spare respiratory capacity, nonmitochondrial respiration and coupling efficiency, were calculated from OCR curves. Results are expressed as mean OCR (pmol/min/μg protein).

### 2.10. Mitochondrial Respiratory Chain Complexes Activity

PC-3 cells were seeded on 100 mm plates and treated with 10 μM of CAVPENET for 48 h in FBS-free RPMI-1640 medium at 37 °C and 5% CO_2_ atmosphere. After 48 h, cells were collected, and the freshly obtained pellets were homogenized using a Glass-Teflon Potter Elvehjem in 2 mL of ice-cold buffer (130 mM sucrose, 50 mM KCl, 5 mM MgCl_2_, 5 mM KH_2_PO_4_, 5 mM HEPES; pH 7.4). Then, after centrifugation of cell homogenates at 1000 g for 10 min at 4 °C, the supernatant (containing cytosolic organelles) was collected and centrifuged at 12,000× *g* for 15 min at 4 °C. The resulting pellet (mitochondria-rich fraction) was re-suspended in washing buffer (250 mM sucrose, 5 mM HEPES, pH 7.2). After centrifugation at 14,000× *g* for 15 min at 4 °C, the pellet was resuspended with ~200 μL of washing buffer. The protein concentration was determined using bicinchoninic acid assay (BCA; Pierce Thermo Fisher Scientific, Inc., USA).

#### 2.10.1. Citrate Synthase Activity

Citrate synthase activity was evaluated spectrophotometrically by monitoring the reduction of 5,5-dithiobis-2-nitrobenzoic acid (DTNB) by the CoA-SH at 37 °C, in 200 mM Tris-HCl (pH 8.0) and 0.02% Triton X-100. The time-dependent increase in absorbance at 412 nm was measured after adding 10 mM DTNB, 100 mM oxaloacetate and 6.1 mM acethyl-CoA to 10 μg of mitochondrial protein. The results were expressed as DTNB mM/min/mg protein.

#### 2.10.2. Complex I Activity

Complex I activity was measured spectrophotometrically, following the nicotinamide-adenine dinucleotide (NADH) oxidation. The reaction kinetics were conducted at 37 °C and pH 7.5, using 25 mM KH_2_PO_4_ and 5 mM MgCl_2_. A total of 5 μg of mitochondrial protein from each condition (untreated and CAVPENET-treated cells) was added to a solution of 100 mM KCN (complex IV inhibitor), 2 mM antimycin A (complex III inhibitor) and 38.75 mM decylubiquinone. The time-dependent decrease in NADH fluorescence intensity at 450 nm (excitation at 366 nm) in the absence and presence of 500 μM rotenone (complex I inhibitor) was recorded. The enzyme activity was determined by the difference between the slopes in the absence and presence of rotenone and expressed as nmol NADH oxidized/min/mg protein.

#### 2.10.3. Complex II Activity

Like complex I activity, the reaction kinetics of succinate oxidation were performed at 37 °C, pH 7.5, in 25 mM KH_2_PO_4_ buffer, by adding 25 mM DCPIP (exogenous final acceptor of electrons), 100 mM KCN, 2 mM antimycin A, 38.75 mM decylubiquinone and 500 μM rotenone, to 5 μg of mitochondrial protein. The time-dependent decrease in absorbance intensity at 600 nM in the absence and presence of 500 nM oxaloacetate (complex II inhibitor) was determined. The enzyme activity was determined by the difference between the slopes in the absence and presence of oxaloacetate and expressed as nmol DCPIP reduced/min/mg protein.

#### 2.10.4. Complex IV Activity

Complex IV activity was measured by the oxidation of cytochrome c at 37 °C and pH 7.5 in the same reaction buffer by the addition of 2 mM antimycin A and 500 μM rotenone to 5 μg of mitochondrial protein. The time-dependent decrease in absorbance at 550 nm in the absence and presence of 100 mM KCN was assessed. As described for complexes I and II, the enzyme activity was determined by the difference between the slopes in the absence and presence of KCN and expressed as nmol of cytochrome c oxidized/min/mg protein.

### 2.11. Statistical Analysis

Statistical analysis was performed using GraphPad Prism version 8.2.1 (GraphPad Software, San Diego, CA, USA). Data are represented as mean ± standard deviation (SD) of at least three independent experiments. Differences between the experimental groups were determined using the Kruskal–Wallis test followed by the post-hoc Dunn’s test, or using the Mann–Whitney U test when only two experimental groups were compared. A *p*-value < 0.05 was considered statistically significant.

## 3. Results

### 3.1. CAVPENET Bioportides Are Internalized in PCa Cells

The intracellular accumulation of TAMRA-conjugated homologs of CAVPENET and CAVPENET control was evaluated in PCa cells (PC-3 and LnCaP). Both bioportides demonstrated cellular internalization after 1 h of incubation in PC-3 (Figure 1A) and LnCaP (Figure 1B) cells. A similar distribution pattern was observed in both PCa cell lines.

### 3.2. CAVPENET Bioportide Decreases the PCa Cell Viability and Migration Ability

To investigate the effects of CAVPENET bioportides on cell viability and migration, both the PC-3 and LnCaP cells were incubated with different doses of bioportides. After a 24 h incubation of the LnCaP and PC-3 cells with 5, 10 and 20 μM of bioportides, only the latter concentration in LnCaP cells produced a significant reduction in cell viability (Appendix A). Conversely, bioportides significantly decreased the PC-3 cells’ viability after a 48 h incubation, when compared with untreated cells (considered 100% viability), except with 5 μM of CAVPENET control (Figure 2A). CAVPENET bioportide also caused a reduction in the LnCaP cells’ viability at a concentration of 10 and 20 µM, whilst only a higher concentration (20 µM) of CAVPENET control was effective after a 48 h incubation (Figure 2B).

Treatment with 10µM CAVPENET bioportide significantly reduced the migration of both PC-3 and LnCaP cells (PC-3 (48 h): *p* = 0.0005, LnCaP (72 h): *p* < 0.0001; Figure 2C,D). Similarly, incubation with 10 μM CAVPENET control also decreased the migration of LnCaP cells (*p* = 0.0003, Figure 2D). Conversely, no significant alterations were observed after the incubation of PC-3 cells with the CAVPENET control bioportide (Figure 2C). Collectively, these results indicate that both CAVPENET control and CAVPENET impact the viability of PCa cells, whilst the CAVPENET bioportide also negatively influences their migration.

### 3.3. CAVPENET Bioportide Decreases the Phosphorylation of AKT at Ser473

To determine the molecular mechanisms underlying the cellular influence of CAVPENET, the phosphorylation levels of AKT at Ser473 were determined. AKT is one of the best-recognized PP1 substrates, dephosphorylated at Ser473 by PP1 [38,39]. Decreased levels of p(Ser473)-AKT were observed after incubation with the CAVPENET bioportide. While PC-3 cells showed a reduction in AKT phosphorylation after incubation with 10 µM of CAVPENET (*p* = 0.0316; Figure 3A), LnCaP cells reduced p-AKT at 10 and 20 µM of CAVPENET (*p* = 0.0316 and *p* = 0.0135, respectively; Figure 3B). Unexpectedly, the CAVPENET control peptide also seemed to reduce the AKT phosphorylation (at Ser473), although with less intensity (Figure 3A,B).

It is well established that AKT depends on the phosphorylation in Ser473 and Thr308 residues for full activation [40]. Thus, given the alterations in AKT phosphorylation (and consequent activity), the phosphorylation levels of GSK3β at Ser9, a substrate of AKT [41], were determined. Although no significant differences were found in PC-3 cells (Figure 3C), incubation of LnCaP with 20 µM of CAVPENET reduced p(Ser9)-GSK3β levels (*p* = 0.0135; Figure 3D).

### 3.4. Molecular Modeling Suggests That CAVPENET Bioportides Bind to the RVxF-Binding Pocket of PP1

To provide structural insight into the relatively similar biological effects of CAVPENET and CAVPENET control bioportides, an investigation was conducted to understand how PP1 recognized these peptides using the AlphaFold Multimer [30] and MD simulations. AlphaFold predicted PP1 with a high predicted IDDT, presenting only 5% of the residues with IDDT below 90. Conversely, CAVPENET and CAVPENET control peptides presented lower IDDT, with average values of 58.88 or 73.25, respectively. Interestingly, the Val2 and Phe4 of the CAVPENET control, which correspond to the RVxF motif, were the only residues with IDDT above 90. IDDT values for both complexes are shown in Appendix A.

The predicted complexes showed both bioportides bound as part of a beta-sheet domain of PP1, into the RVxF-binding pocket (Figure 4A). When inspecting the interface of the complexes, it was observed that hydrophobic residues located at positions two (valine) and four (phenylalanine for CAVPENET control and isoleucine for CAVPENET) of both peptides are accommodated in a hydrophobic region of PP1, which is formed by I169, L243, F257 and F293 residues (Figure 4A, zoomed regions). These residues present a high (>90%) percentage of contacts when all frames from MD simulations were analyzed. The percentage of contacts per residue for each complex is shown in Appendix A.

The RMSF calculation, considering frames from all three replicas, showed that the range of residues two to five presents the lowest values, indicating that such a region is bound tighter to PP1, while residues at positions above five show unstable interactions with PP1 (Figure 4B). Indeed, the C-terminus of both peptides presents a more diverse set of conformations, with some of them moving towards the solvent (Appendix A).

Another similarity between the complexes was regarding the main chain atom interactions. The residue C291 of PP1 was essential to stabilize hydrogen bonds between the backbone atoms of both peptides at positions three and five. To analyze this, the pairwise distance of the following backbone atoms was calculated: the hydrogen of the backbone nitrogen (HN) and the backbone oxygen (O) of those three residues. The distribution of the distances between C291 of PP1 and CAVPENET control (D3 or I5) or CAVPENET (K3 or D5) residues showed that they are kept in contact at short distances, below 2.5 Å. These interactions can be observed in the structure of each complex in Appendix A.

### 3.5. CAVPENET-Induced PCa Cell Death Depends on PP1 Activity in an Isoform-Dependent Manner

To investigate the potential involvement of PP1 in CAVPENET-induced cell death, PP1α phosphorylation levels at Thr320 were evaluated. The phosphorylation of PP1α at this residue is recognized as a mechanism of PP1 inhibition [38]. Incubation with CAVPENET resulted in decreased levels of p(Thr320)-PP1α, suggesting increased PP1α activity (Figure 5A). Notably, since this Thr residue is conserved across all PP1 isoforms (at position 320 in PP1α, 316 in PP1β and 311 in PP1γ), the results suggest that overall PP1 activity might be increased [42,43]. Furthermore, to explore the potential isoform-dependent modulation of PP1 activity by the CAVPENET bioportide, the viability of the PCa cells after knockdown of PP1α, PP1β and PP1γ alone or in combination with CAVPENET was assessed. The success of the siRNA knockdown was confirmed for all three PP1 isoforms (Figure 5B,D,F). The PP1α and PP1γ knockdowns did not significantly affect the PC-3 cells’ viability. The CAVPENET-induced cell death appears to not depend on PP1α isoform since decreased expression of PP1α in cells treated with CAVPENET did not recover their viability (Figure 5C). On the other hand, CAVPENET-induced decreased viability was recovered when combined CAVPENET treatment with PP1γ knockdown (*p* = 0.0002, when compared CAVPENET with siRNA *PPP1CC* + CAVPENET) (Figure 5G). Regarding PP1β, its knockdown significantly decreased the PC-3 cells’ viability (*p* < 0.0001). When combined with the CAVPENET peptide, a stronger decrease in the cells’ viability was observed (*p* = 0.0029, when compared to CAVPENET with siRNA *PP1CB* + CAVPENET) (Figure 5E).

The CAV1 protein interacts with both PP1 and PP2A through its scaffolding binding domain, resulting in the suppression of their phosphatase activity, thereby promoting the phosphorylation of AKT at Ser 473 [7]. Interestingly, the presence of a PP1-binding motif situated upstream of the scaffolding domain of CAV1 (^65^KIDF^68^) was identified, which appears to contribute exclusively to PP1 binding (Figure 5H) To delve further into the contribution of PP2A on CAVPENET-induced reduction in the PC-3 cells’ viability, simultaneous incubation of PC-3 cells with CAVPENET and a potent PP2A inhibitor, cantharidin [44], was conducted. The concentration of cantharidin was selected based on the literature [45]. Notably, the incubation of PCa cells with CAVPENET + cantharidin did not recover cell viability (Figure 5I), suggesting that PP2A was not involved in the observed effect of CAVPENET. To gain deeper insights into the ability of CAVPENET to bind PP2A, the same molecular modeling protocol as for PP1 was employed. AlphaFold predicted lower IDDT values when the peptides were bound to PP2A (Appendix A). Moreover, peptides were predicted to bind to the ion binding site. Indeed, the binding sites in PP1 and PP2A present distinct residue compositions, in which F257 and C291 residues of PP1 are substituted by W250 and Y284 in PP2A (Appendix A). These substitutions may affect the physiochemical properties of the RVxF-binding pocket, likely explaining the specificity of the bioportides synthesized in our work. Altogether, these results infer that CAVPENET-induced PCa cell death primarily depends on PP1 modulation rather than PP2A. This effect seems to depend on the PP1γ isoform, but not PP1α and β.

### 3.6. Modulation of PP1 Activity by CAVPENET Bioportide Results in Suppression of PCa Cells Glycolytic Metabolism

As previously mentioned, PP1 is associated with AKT inhibition, and several authors have proposed a role for AKT in the regulation of cancer cell metabolism [46,47,48]. To gain a deeper insight into the impact of CAVPENET-induced AKT inhibition on the metabolic pathways of PCa cells, the oxygen consumption rate (OCR) and extracellular acidification rate (ECAR) were monitored over a span of 100 min, measured every 8.5 min. This involved a series of injections, including oligomycin (an ATP synthase inhibitor), carbonyl cyanide-4 (trifluoromethoxy) phenylhydrazone ((FCCP) as an uncoupling agent) and a combination of rotenone and antimycin A, acting as inhibitors of complex I and III, respectively [49]. Though no significant alterations were observed in the OCR of the PCa cells treated with 10 μM of CAVPENET (Figure 6A), proton leak decreased in CAVPENET-treated cells (*p* = 0.0159). Nonetheless, no significant alterations were found for the remaining OCR parameters (Figure 6B). The involvement of eukaryotic translation initiation factor 4E (eIF4E)-binding protein 1 (4E-BP1) in AKT regulation of mitochondrial respiratory capacity has been proposed [46]. Contrary to expectations, neither the phosphorylation of 4E-BP1 at Thr37/46 (Figure 6C) nor the activity of mitochondrial respiratory complexes (Figure 6D) were significantly altered in CAVPENET-treated cells, despite the lower activity of AKT caused by CAVPENET peptide.

Regarding ECAR, decreased levels before (*p* = 0.0276) and after (*p* = 0.0050) injection of oligomycin were noted in PCa cells incubated with CAVPENET (Figure 6E,F). The injection of oligomycin blocks ATP production through oxidative phosphorylation (OXPHOS), thus driving cells to use glycolysis at its maximum capacity [49,50]. These results suggested an impairment of the glycolytic pathway in peptide-treated PCa cells. To gain further insight into the blockage of the glycolytic pathway, the expression of hexokinase 2 (HK2) was evaluated, as an interplay was previously reported between HK2 and AKT [47,51]. Indeed, decreased levels of HK2 were observed after incubation of PCa cells with 10 μM of CAVPENET (*p* = 0.0286, Figure 6G). Altogether, these findings suggest that CAVPENET decreased the glycolytic activity of PCa cells.

### 3.7. CAVPENET Bioportide Modulates the Lipid Metabolism of PCa Cells

Given the observation that CAVPENET causes a blockage in glycolysis, it was hypothesized that the PCa cells could regulate their metabolic behavior to compensate for decreased glycolytic flux. In this context, the protein expression of several key regulatory enzymes of fatty acid metabolism was analyzed (Figure 7) to provide insight into the flux through pathways. Thus, the observed reduced levels of fatty acid synthase (FASN) (*p* = 0.0286), the rate-limiting enzyme of the fatty acid synthesis pathway, in CAVPENET-treated cells suggested impairment of fatty acid synthesis (Figure 7A). A reduced expression of CD36, a transmembrane protein involved in fatty acids uptake, was also noted (*p* = 0.0286, Figure 7B). Moreover, the translocation of fatty acids through the mitochondria, by carnitine palmitoyltransferase (CPT1) is considered the limiting step of fatty acids oxidation. Contrary to what was verified for FASN and CD36 expression, CAVPENET increased the expression levels of CPT1 (*p* = 0.0286, Figure 7C).

To evaluate how increased fatty acid oxidation contributes to CAVPENET anticancer activity, a well-known inhibitor of CPT1 was used. The concentration of etomoxir was selected based on previous studies conducted in cancer cells [52,53]. The incubation of PCa cells with CAVPENET + etomoxir did not affect the cells’ viability compared with CAVENET alone (*p* < 0.0001, for both, when compared with untreated cells) (Figure 7D).

Overall, these results suggested that incubation of PCa cells with the CAVPENET bioportide increases lipids catabolism but decreases both lipid synthesis and uptake.

## 4. Discussion

Targeting protein phosphorylation has long been explored as a strategy for cancer treatment. For instance, several kinase inhibitors have been approved by the FDA as anticancer therapies [54]. In the context of phosphatases, targeting the PP1 active site was one of the earliest proposed approaches for anticancer therapeutic development. The potential therapeutic effect of PP1 inhibitors tautomycin and calyculin A was evaluated in cancer cells. Despite their effectiveness in causing cancer cell apoptosis, both were associated with significant toxic effects, limiting their applicability [55,56]. More recently emerged the concept of target-specific PP1 complexes, as a more selective approach, allowing the modulation of PP1 activity against specific substrates [6]. The CAV1 protein is one of the best characterized PP1 interactors in the context of PCa [57]. Indeed, CAV1 has been associated with tumor progression in PCa and the CAV1-induced inhibition of PP1 activity seems to contribute to its tumor promoter role, highlighting the potential of this complex as a drug target [7,58]. In this study, a bioportide mimicking the PP1-binding motif of CAV1 was synthesized, by employing microwave-assisted solid-phase peptide synthesis. A combination of molecular modeling and biological investigations unraveled the intricate molecular and cellular impacts of this bioportide on PCa cells. Our findings underscore the significance of harnessing peptides as a potent anticancer strategy for PCa.

The bioportide was designed based on an initial bioinformatic analysis of the CAV1 sequence, using the ScanProsite online tool. This exploration revealed the presence of a PP1-docking motif (RVxF motif)—^65^KIDF^68^. Thus, the CAVPENET bioportide was synthesized as a tandem, sychnologic construct with the penetratin CPP at the C-terminal. A control bioportide with a scrambled docking motif (CAVPENET control) was also prepared and characterized. Both bioportides were successfully internalized in androgen-dependent (LnCaP) and castration-resistant (PC-3) PCa cells. CPPs are widely recognized for their ability to cross the plasma membrane and deliver otherwise impermeable cargoes into cells, enabling these cargoes to reach their molecular targets across various cell types. However, the molecular mechanisms involved in such processes are not fully understood. Direct membrane translocation and/or endocytosis have been described as the main general mechanisms [59,60]. Penetratin was one of the first CPPs discovered, and it is widely recognized by its ability to internalize into cells, including cancer cells, and to improve the intracellular delivery of anticancer agents [26,61]. Penetratin has been widely used to promote cellular uptake also due to its absence of toxic effects observed in several types of cells, including different tumor cell lines, even in high concentrations [61,62].

The tumor promoter role of CAV1 is now well-established in PCa. In fact, different CAV1-based anticancer therapies have been proposed for several tumor types [63,64]. Additionally, the efficacy of peptides based on the CAV1 scaffolding domain (^82^DGIWKASFTTFTVTKYWFYR^101^) in impairing in vitro cancer cell migration [65] and delaying in vivo tumor progression were reported [66]. Although this CAV1 region was proposed by Li et al. (2003) [7] as responsible for the interaction with PP1, its involvement in the interaction of CAV1 with a wide range of other signaling proteins was reported, including PP2A, G-proteins and Src-like kinases [67]. Herein, a search for PP1-binding motifs from the CAV1 sequence found another region crucial for the interactions with PP1, located upstream of the CAV1 scaffolding domain. It was observed that the incubation of LnCaP and PC-3 cells with the CAVPENET bioportide (designed to mimic the identified PP1-binding motif of CAV1) decreased both the viability and migration ability of PCa cells in a concentration-dependent manner.

Mechanistically, the PP1/PP2A-dependent inhibition of protein kinase B (AKT) signaling was associated with the effect of CAV1 scaffolding domain-based peptides. In addition, aberrant expression and/or activity of AKT has been considered a hallmark of cancer associated with the survival and invasion of PCa cells [7,68]. Although AKT signaling has been thoroughly investigated as a promising target for cancer therapy, with several peptide AKT inhibitors documented in the literature and some of them approved by the FDA for cancer treatment [69,70,71,72], its exploration in PCa remains scarce. Moreover, the AKT-targeting peptides described in the literature directly target AKT or the interaction with co-activators or substrates. Here, we propose an indirect targeting of AKT, by specifically modulating the activity of PP1. Indeed, CAVPENET was able to increase PP1 activity, decreasing the phosphorylation levels of AKT at Ser473, and thus reducing its activity [40] in both PC-3 and LnCaP cells. Since GSK3β is a downstream target for activated AKT, its phosphorylation levels were also determined. Despite the obvious alterations in phosphorylation of AKT in both PCa cell lines, no significant differences were apparent concerning GSK3β phosphorylation levels in peptide-treated cells, except for LnCaP cells exposed to the higher concentration of CAVPENET. A reduction in phosphorylation of GSK3β was expected either directly through increased activity of PP1 [73], which dephosphorylates GSK3β at Ser9, or indirectly by PP1-induced decreased activity of AKT, which is capable of phosphorylating GSK3β at the same residue [41]. The phosphorylation of downstream targets, including GSK3β, has been proposed as a major player in AKT-induced cancer cell survival [74]. Yet, it appears that the reduced viability and migration of PCa cells induced by CAVPENET is not linked to the dephosphorylation of GSK3.

Since the CAVPENET control bioportide sequence contains a scrambled PP1-binding motif, it was anticipated that this peptide would display negligible biological activity. However, following effective internalization, the CAVPENET control peptide likewise reduced the viability and migration of PCa cells associated with attenuated AKT phosphorylation These similar activities may be the consequence of a common molecular mechanism in which both bioportides bind the same region of PP1. Despite the sequence differences, both bioportides have hydrophobic amino acids located in the same positions (two and four). These residues were identified, using RMSF calculations, as major contributors to PP1 binding site accommodation, forming hydrogen bonds between the backbone atoms of the complex. Such positions correspond to the Val and Phe residues of the canonical RVxF motif. Previous structural works have shown that the substitution of phenylalanine for other hydrophobic residues, such as tryptophan or leucine, does not prevent peptide binding to PP1 [75,76]. Therefore, the substitution of Phe by Ile may display no or small effects in peptide binding to PP1.

The involvement of PP1 in CAVPENET-induced cell death was confirmed by targeted suppression of the PP1 isoform expression using siRNAs. Indeed, as mentioned above, the PP1 family includes different isoforms, PP1α, β and γ, all of which are expressed in both LnCaP and PC-3 cells [43]. Though little is known about the PP1 isoform-specific interactors, a preference for specific PP1 isoforms has been described for certain RIPPOs [77,78,79]. Moreover, the action of some compounds, such as prodigiosin, appears to be PP1-isoform specific, potentially impacting the signaling pathways affected by that specific isoform [80]. Although the Human Integrated Protein–Protein Interaction rEference (HIPPIE) database describes CAV1 as an interactor of PP1α with a score of 0.78, no evidence of the PP1 isoform involved is currently available [7,81]. The results of the present study indicate PP1γ to be the major isoform involved in the antitumoral effect of CAVPENET. Indeed, reduced expression of *PPP1CC*, but not *PPP1CA* and *PPP1CB*, almost fully recovered the viability of CAVPENET-treated PCa cells.

The possible mechanistic link between CAVPENET modulation of PP1 activity and metabolic changes in PCa cells was further evaluated. The relevance of glycolytic metabolism for PCa development and progression is widely recognized [82]. Furthermore, it has been proposed that the hyperactivation of AKT and the resulting increase in glycolytic metabolism provide advantages for the growth of tumor cells [83]. The involvement of AKT-induced HK2 expression and activity in tumor growth was reported in different types of cancer [84,85,86], including in PCa [87]. For instance, several compounds, including Glycyrrhizin [85], Licochalcone A [88] and σ-tocotrienol [47], have been successfully used to suppress cancer cell proliferation by decreasing HK2 expression via AKT signaling. Likewise, this mechanism would explain the detrimental impact of CAVPENET on proliferation and migration ability, since decreased glycolytic metabolism and expression of HK2 were observed in CAVPENET-treated PCa cells.

The inhibition of AKT was also associated with an adverse influence on mitochondrial respiration. Indeed, decreased AKT activity correlates with decreased OCR in different malignancies [89,90]. The 4E-BP1-dependent downregulation of respiratory complexes and inhibition of pyruvate dehydrogenase (PDH) were already implicated in the latter [46,90]. Though no significant alterations were determined in OCR parameters, a general tendency of OCR decrease in CAVPENET-treated cells was observed. However, CAVPENET had no noticeable impact on the 4E-BP1 phosphorylation levels and mitochondrial complexes’ activity, suggesting no significant impact on mitochondrial respiration.

Because the CAVPENET bioportide did not significantly change mitochondrial metabolism in PCa cells, it was hypothesized that CAVPENET treatment may enhance cellular lipid metabolism as a compensatory response to glycolysis inhibition. In fact, lipid metabolism plays a major role in PCa cells [91] and a relationship between cancer-associated AKT hyperactivation and lipid metabolism deregulation has been proposed [92]. Along with increased glycolytic metabolism, AKT also drives synthetic pathways contributing to cancer cell growth. Indeed, the AKT enhances lipid availability to supply the aberrant cancer cellular growth, by suppressing lipid catabolism and increasing their synthesis [92,93,94]. Several authors described a positive association between free fatty acid availability and PCa cell proliferation and migration [95,96,97,98]. The palmitoylation of proteins, including WNT1, was proposed as the main oncogenicity mechanism of fatty acids [98]. Contrarily, Zhu et al. (2021) [99] suggested an anti-tumor role for palmitic acid in PCa cells. An enhanced expression of CPT1 in CAVPENET-treated PCa cells was found, suggesting increased fatty acid oxidation, which is consistent with the CAVPENET-induced AKT inhibition. Decreased fatty acid synthesis, through underexpression of FASN, also contributed to CAVPENET-induced suppression of fatty acid availability. Blocking FASN was associated with increased CD36 expression, ensuring the cellular availability of fatty acids to sustain cancer cell proliferation [100]. Nevertheless, CAVPENET-induced downregulation of FASN was accompanied by diminished CD36 expression, underscoring the reduced availability of fatty acids in CAVPENET-treated PCa cells. Research in cancer cells revealed a positive association between CD36 and AKT [101], suggesting that CAVPENET inhibition of AKT might be responsible for reducing the availability of fatty acids in PCa cells, via distinct mechanisms.

Despite the apparent anticancer role of CPT1, by reducing the availability of fatty acids, excess CPT1 seems to support PCa cell proliferation by an ROS-mediated stress phenotype [102]. Moreover, a CPT1 knockout was associated with inhibition of PCa cell invasion [103], becoming an important therapeutic target for PCa. In this context, different compounds have been produced to target this protein [52]. Etomoxir, one of the best known CPT1 inhibitors, was evaluated in PCa patients during a phase II clinical trial; however, hepatotoxicity hampered definitive testing [104]. Alternatively, other drugs targeting CPT1A, such as Ranolazine and Perhexiline, showed efficacy when combined with anticancer drugs (e.g., Enzalutamide) [103]. Hence, the combination of CAVPENET with a CPT1A inhibitor was investigated as an approach to improve the anticancer activity of CAVPENET. However, the co-administration of CAVPENET and etomoxir did not further promote the CAVPENET-induced decrease in PCa cell viability.

In addition to its demonstrated potential in treating PCa, CAVPENET may also be applicable to other cancer types. For instance, the target complex PP1/CAV1 has already been identified in colorectal cancer cells [105], suggesting that CAVPENET could be directly applicable to colorectal cancer treatment. Moreover, the approach outlined in this study could be adapted to target other PP1 complexes that play critical roles in the progression of various cancers.

## 5. Conclusions

Collectively, our data demonstrate that the CAVPENET bioportide is internalized by PCa cells and modulates PP1γ activity, decreasing AKT phosphorylation. Consequently, AKT inhibition might block glycolytic metabolism and alter lipid metabolism, impairing PCa cell proliferation and migration (Figure 8). This approach allows for a more selective modulation of AKT signaling, compared with others reported in the literature. In addition, to minimize toxic effects, it potentially enables a tissue-specific effect. Thus, our findings emphasize the potential of the designed peptide as an innovative therapy for PCa, paving the way for further evaluation in a more sophisticated model to fully develop its therapeutic potential.

## Figures and Tables

**Figure 1 pharmaceutics-16-01199-f001:**
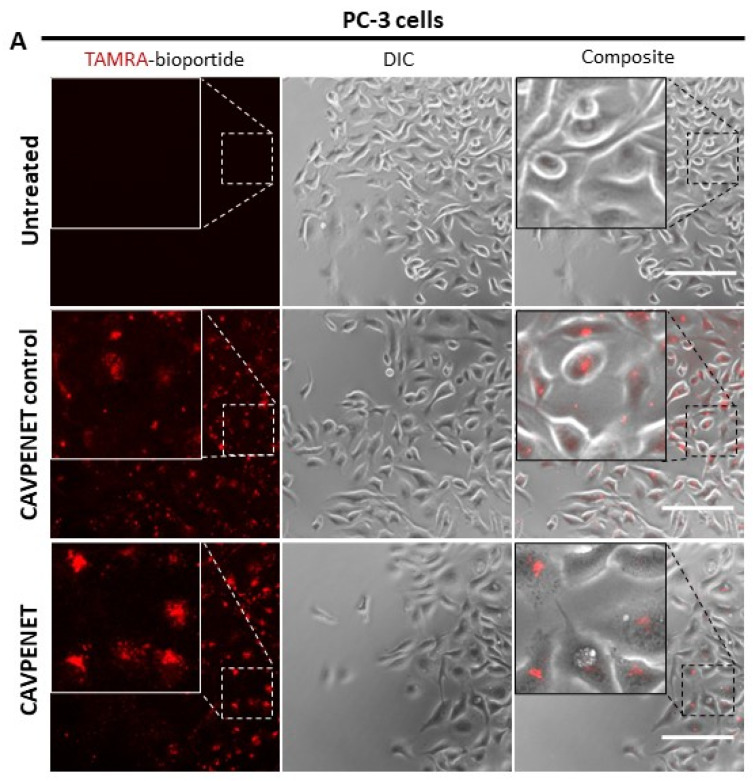
Translocation of CAVPENET control and CAVPENET bioportides into (**A**) PC-3 and (**B**) LnCaP cells. PCa cells (PC-3 and LnCaP) were incubated with TAMRA-labeled bioportides (5 µM) for 1 h at 37 °C and 5% CO_2_ atmosphere. Representative images from 3 independent experiments are represented. DIC: differential interference contrast. Scale bar: 150 μm.

**Figure 2 pharmaceutics-16-01199-f002:**
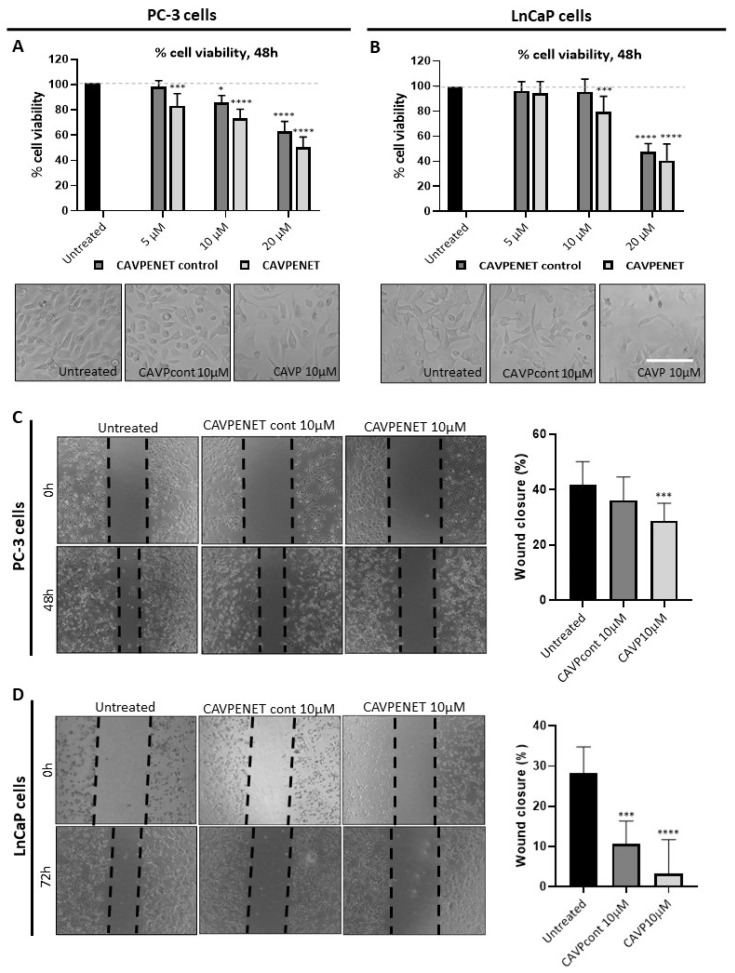
Cell viability and migration ability of PC-3 ((**A**,**C**), respectively) and LnCaP ((**B**,**D**), respectively) cells, after incubation with CAVPENET control and CAVPENET bioportides. The cells were incubated with bioportides (5, 10 and 20 µM) for 48 h and the cells’ viability was evaluated using PrestoBlue cell viability assay. The percentage of cell viability was calculated through the ratio between treated and untreated conditions, considering the untreated condition as 100% viability. The results are expressed as mean ± SD from four independent experiments with five replicates/condition. The migration was evaluated by the wound healing assay. Five measures per replicate for each timepoint were performed, and the results are represented as mean ± SD. * *p* < 0.05; *** *p* < 0.001; **** *p* < 0.0001. Scale bar: 150 μm.

**Figure 3 pharmaceutics-16-01199-f003:**
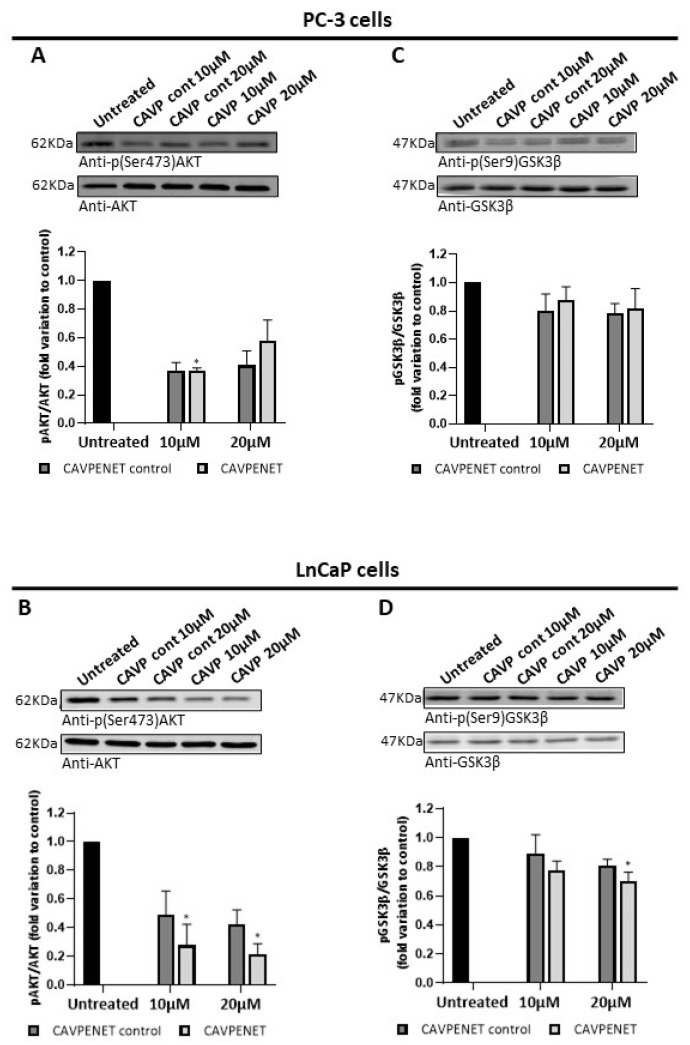
Protein expression levels of (p)-AKT (Ser473) and (p)-GSK3β (Ser9) in PC-3 ((**A**,**C**), respectively) and LnCaP ((**B**,**D**), respectively) cells after treatment with the CAVPENET control and CAVPENET bioportides. The cells were incubated with bioportides (10 and 20 µM) for 48 h, collected and lysed using RIPA1x lysis buffer. The proteins were resolved by SDS-PAGE and transferred to membranes, that were incubated with the respective antibodies. The band’s intensity was quantified using ImageLab software, normalized to PonceauS staining intensity and represented as fold change to control (untreated cells). The results are expressed as mean ± SD from three independent replicates. * *p* < 0.05.

**Figure 4 pharmaceutics-16-01199-f004:**
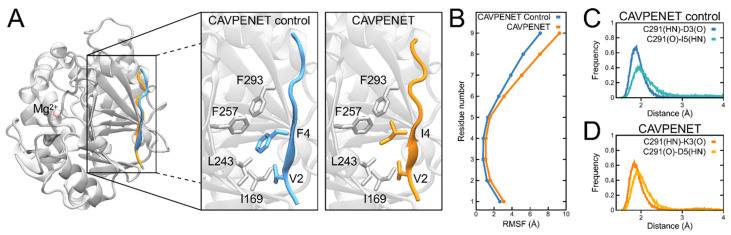
Structural aspects of the interaction between PP1 and CAVPENET bioportides. (**A**) The predicted structure of PP1 (gray) shows CAVPENET control (cyan) and CAVPENET (orange) bioportides bound in the same region. Mg^2+^ (pink sphere) is located in the RVxF-binding pocket of PP1. Both peptides interact in a hydrophobic region of PP1, composed of I169, L243, F257 and F293 residues (zoomed panels). (**B**) RMSF calculation per residue of each bioportide shows low fluctuations in the range of 2–4 residues and increasing fluctuations from residue 5 and above. The frequency of the distance of backbone residues of PP1 (C291) and (**C**) CAVPENET control (D3 and I5) and (**D**) CAVPENET bioportides (K3 and D5) suggests that the complex is also stabilized by hydrogen bonds.

**Figure 5 pharmaceutics-16-01199-f005:**
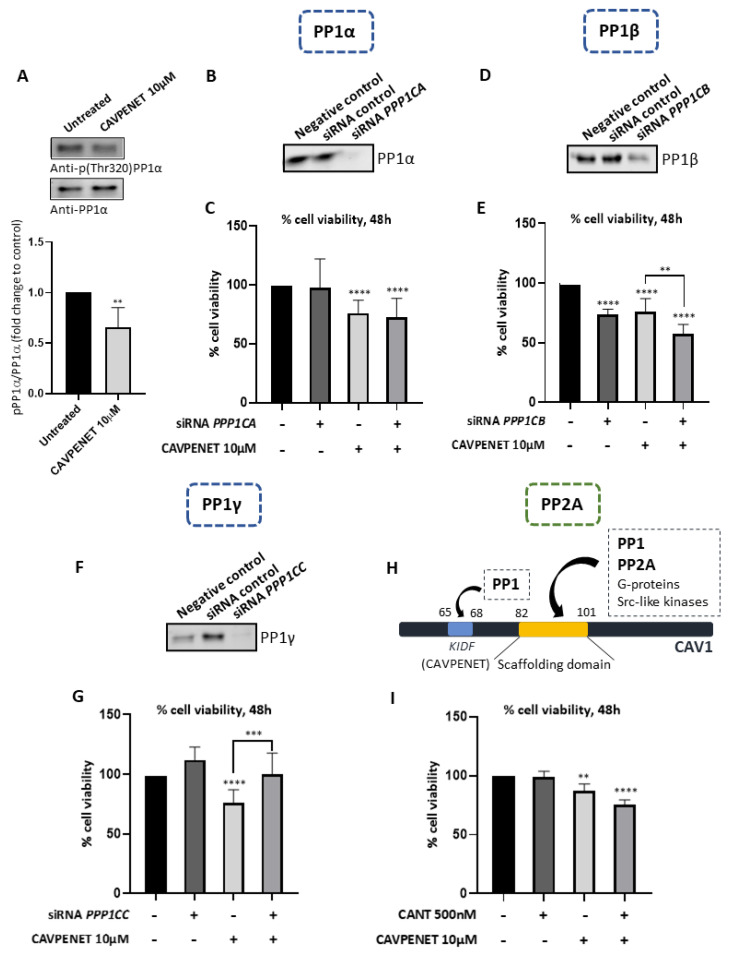
Protein expression levels of (p)-PP1α (Thr320) (**A**). Cell viability of siRNA PP1CA (**C**), siRNA PP1CB (**E**), siRNA PP1CC (**G**) and 500 nM of PP2A inhibitor cantharidin (**I**), with CAVPENET 10 μM and the respective combinations after 48 h of incubation. The reduced expression of PP1α (**B**), β (**D**) and γ (**F**) after the respective siRNA knockdown is also represented. A schematic representation of PP1 and PP2A-binding to CAV1 is represented in (**H**). The cells were treated as previously described and the cells’ viability and protein expression were measured. The results were expressed as mean ± SD from at least three independent replicates, with the untreated cells considered 100% viability. ** *p* < 0.01, *** *p* < 0.001, **** *p* < 0.0001.

**Figure 6 pharmaceutics-16-01199-f006:**
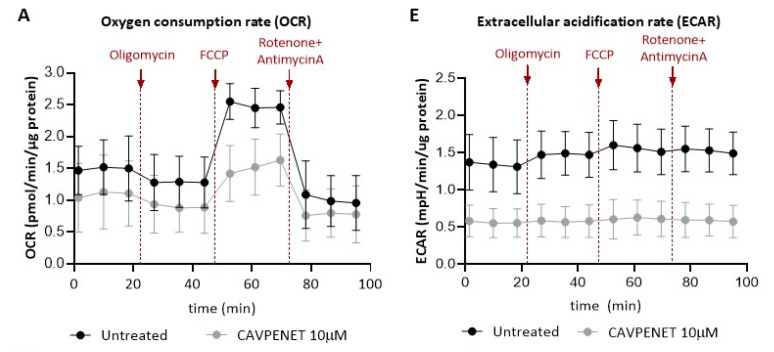
Oxygen consumption rate (OCR) (**A**) and the respective parameters (Basal respiration, proton leak, maximal respiration, spare respiratory capacity, non-mitochondrial respiration and ATP-production coupled respiration) (**B**); protein expression levels of (p)-4E-BP1 (Thr37/46) (**C**); activity of citrate synthase and mitochondrial complexes I, II and IV (**D**); extracellular acidification rate (ECAR) (**E**) and respective parameters (**F**); and protein expression levels of hexokinase 2 (HK2) (**G**) after treatment with CAVPENET. For the measurement of OCR and ECAR, the cells were incubated with 10 μM CAVPENET bioportide for 48 h and these parameters were evaluated along the time using the Agilent Seahorse XF Cell Mito Stress Test Kit. For the determination of (p)-4E-BP1 and HK2 expression, the cells were treated as previously described and lysed and resolved using the same approach. The activity of the mitochondrial complexes was determined using spectrophotometric enzymatic assays, as described in detail in the methods section. All the results are expressed as mean ± SD from four independent experiments. * *p* < 0.05, ** *p* < 0.01.

**Figure 7 pharmaceutics-16-01199-f007:**
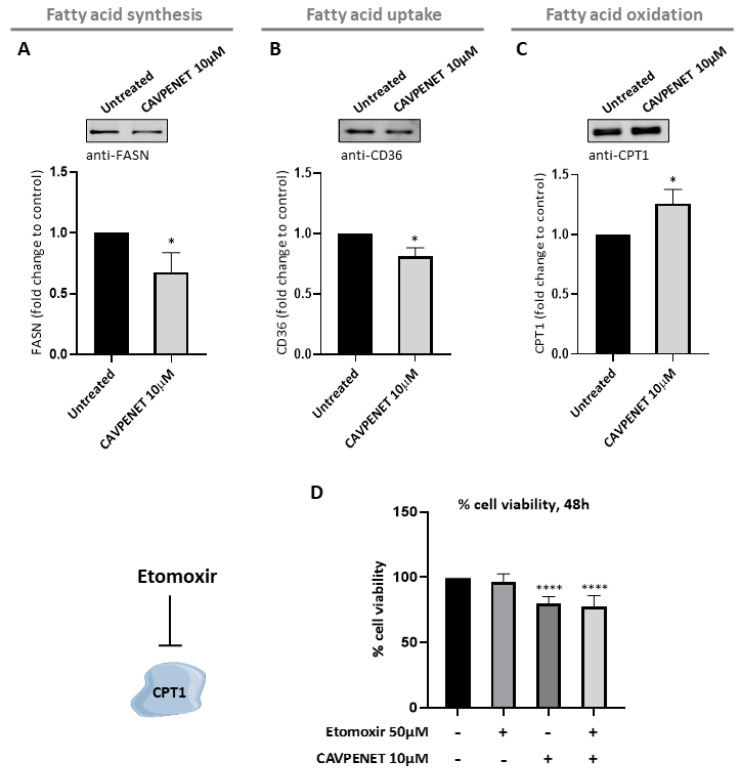
Protein expression levels of fatty acid synthase (FASN) (**A**), CD36 (**B**) and carnitine palmitoyltransferase (CPT1) (**C**) after incubation with CAVPENET. Cell viability of PC-3 cells after 48 h incubation with 50 μM of etomoxir (CPT1 inhibitor), 10 μM of CAVPENET and their combination (**D**). The PC-3 cells were incubated with 10 µM of CAVPENET for 48 h, collected and lysed using RIPA1x lysis buffer. The proteins were resolved by SDS-PAGE and transferred to membranes, that were incubated with the respective antibodies. The band’s intensity was quantified using ImageLab software, normalized to PonceauS staining intensity and represented as fold change to control (untreated cells). The cell viability was determined using PrestoBlue cell viability reagent assay, considering untreated cells as 100% viability. The results are expressed as mean ± SD from at least three replicates. * *p* < 0.05; **** *p* < 0.0001.

**Figure 8 pharmaceutics-16-01199-f008:**
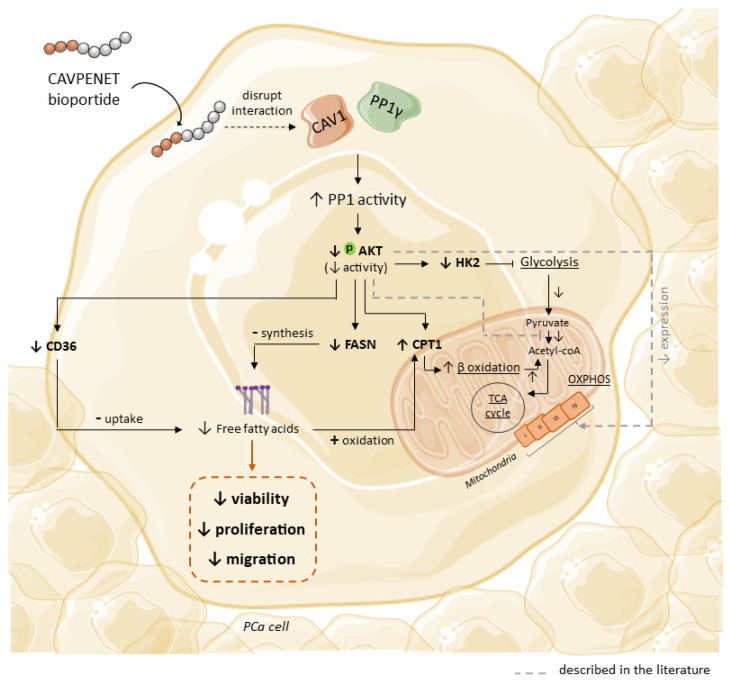
Proposed molecular mechanisms that drive the effect of CAVPENET bioportide. Briefly, CAVPENET is taken up by the prostate cancer cells and increases the activity of PP1, probably by disrupting the interaction with CAV1. The increased activity of this phosphatase decreases the phosphorylation and activity of AKT, ultimately leading to blockage of glycolysis, by decreasing HK2 expression, and decreases fatty acid synthesis and uptake, by decreasing FASN and CD36 expression. To compensate for these alterations, PCa cells increase lipid oxidation by increasing CPT1 expression. The alterations observed in this work are highlighted in bold.

**Table 1 pharmaceutics-16-01199-t001:** Designation, sequence and masses of the bioportides.

Bioportides Designation	Sequence	Mass (g/mol)
CAVPENET	VV**KIDF**EDRQIKIWFQNRRMKWKK	3191.84
CAVPENET control	VV**DFIK**EDRQIKIWFQNRRMKWKK	3191.84

NOTE: The PP1 binding motif is in **bold**, and the cell-penetrating peptide (penetratin) is underlined. Both peptides were synthesized as C-terminal amides.

## Data Availability

All data are available within the manuscript and its Appendix A.

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
