# Peer review of "CAVPENET Peptide Inhibits Prostate Cancer Cells Proliferation and Migration through PP1γ-Dependent Inhibition of AKT Signaling"

_pharmaceutics, 2024, doi:10.3390/pharmaceutics16091199_

Round 1

Reviewer 1 Report

Comments and Suggestions for Authors

Dear Authors,

Please see attached my report.

Thanks

Author Response

 The article by Matos et al., entitled CAVPENET peptide inhibits prostate cancer cells proliferation and migration through PP1γ-dependent inhibition of AKT signaling deals with a designed peptide and its analogue that can mimic PP1 docking motif to stop prostate cancer progression. While the study is well designed, conducted and concluded, and it will add to the toolbox of potential peptide therapeutics, I have the following comments before accepting the article for publication.

Major comments

Comment 1: It would be impactful the authors dedicate a paragraph or two about the FDA approved peptide-based therapeutics that targets PCa. Maybe also highlight the differences in the mechanism of action for the one used as therapy (Lutetium (177Lu)) and their new CAVPENET peptide. 68Ga-PSMA-11: Diagnosis of recurrent prostate carcinoma.

  • Piflufolastat-F18 (PylarifyTM): Positron emission tomography (PET) of prostate-specific membrane antigen (PSMA)-positive lesions in men with prostate cancer
  • Lutetium (177Lu) vipivotide tetraxetan (PluvictoTM): To treat prostate-specific membrane antigen-positive metastatic castration-resistant prostate cancer after other therapies.
  • Flotufolastat F-18 (PoslumaTM): To use with positron emission tomography (PET) imaging in certain patients with prostate cancer.

Response 1: Based on the reviewer suggestion, the following information was added in the Introduction section of the manuscript: “Peptide therapeutics have gained significant recognition in recent years. A notable example in the treatment of PCa, is the peptide-based therapeutic agent 177Lu-vipivotide terraxetan, recently approved by FDA. This anticancer drug selectively targets the prostate-specific membrane antigen (PSMA) and delivers beta-radiations to effectively destroy prostate cancer cells, underscoring the potential of peptide-based therapies in oncology”

Comment 2: It is very important to include a paragraph that describes the action of TKIs and PPI. Where the former inhibit phosphorylation and the latter boost desphosphorylation. Maybe this also could be stated at the end of the article as future perspectives and directions.

Response 2: Based on the reviewer comment, we added a better context for targeting phosphorylation process, including some information about kinase inhibitors, in the beginning of the Discussion section: “Targeting protein phosphorylation has long been explored as a strategy for cancer treatment. For instance, several kinase inhibitors have been approved by the FDA as anticancer therapies55. In the context of phosphatases, targeting the PP1 active site was one of the earliest proposed approaches for anticancer therapeutic development. The potential therapeutic effect of PP1 inhibitors tautomycin and calyculin A was evaluated in cancer cells. Despite effective in causing cancer cells apoptosis, both were associated with significant toxic effect, limiting their applicability56,57. More recently, emerged the concept of target specific PP1 complexes as a more selective approach, allowing the modulation of PP1 activity against specific substrates”

Comment 3: It is worth adding more discussion about CPP, where the peptide in this study is delivered using this valuable class.

Response 3: The value of CPPs as a drug delivery technology was highlighted in the introduction section. According with the reviewer suggestion, we also highlighted its potential in discussion section improving the following text: “CPPs are widely recognized for their ability to cross the plasma membrane and deliver otherwise impermeable cargoes into cells, enabling these cargoes to reach their molecular targets across various cell types.”. Moreover, we include some information about the particular CPP used in this study, penetratin: “Penetratin was one of the first CPPs discovered, and it is widely recognized by its ability to internalize into cells, including cancer cells and to improve the intracellular delivery of anticancer agents”

Comment 4: Please include the LCMS chromatograms for all peptides (including with fluoro tag).

Response 4: The peptides described in the manuscript were all purified using semi-preparative scale HPLC equipment, specifically dedicated to their purification employing extended water/acetonitrile gradients to facilitate rigorous separation of final products from all impurities. However, due to the recent reorganization of RIHS at Wolverhampton University, and the consequent decommissioning of the peptide synthesis laboratory, the retrieval of appropriate chromatograms is not feasible.

Comment 5: Have the peptides checked with the CPP vehicle (penetratin)? Any data to report, even if negative data such as minimal or no internalisation was noticed?

Response 5: The internalization of both peptides was evaluated (Fig.1), but not compared with the CPP vehicle (penetratin) alone. Nevertheless, the ability of penetratin to internalize into cells, including cancer cells is widely recognized, including its ability to improve the intracellular delivery of anticancer agents. This information was added in Discussion section: “Penetratin was one of the first CPPs discovered, and it is widely recognized by its ability to internalize into cells, including cancer cells and to improve the intracellular delivery of anticancer agents25,59. Penetratin has been widely used to promote cellular uptake also due to its absence of toxic effects observed in several types of cell, including different tumor cell lines, even in high concentrations60,61.

Comment 6: Any toxicity studies regarding your peptide?

Response 6: We did not evaluate the effect of the peptide on “normal” cells in this study. However, in future research focused on developing CAVPENET into a drug-like peptide, we may explore incorporating a homing peptide to specifically target prostate cancer cells. At that stage, conducting toxicity studies will be essential.

Furthermore, we are currently assessing the effect of the CAVPENET peptide in a more advanced 3D spheroid model. Specifically, we tested the impact of CAVPENET and its control at concentrations up to 20µM on 3D spheroids composed exclusively of myofibroblasts (WPMY cells) and cancer-associated fibroblasts (CAFs). Preliminary results suggest that these peptides do not significant affect the viability of these cells.

Comment 7: Have you obtained ethical approval for the human cancer cell lines?

Response 7: Thank you for your question regarding the ethical approval for the use of human cancer cell lines in our study. The cell lines used in our research are commercially available immortalized human cancer cell lines (PC-3 and LnCaP), which were obtained from ATCC. These cell lines are widely recognized and utilized within the scientific community.

Furthermore, our research complies with all relevant guidelines and standards set by the Institute of Biomedicine (University of Aveiro) and adheres to the principles outlined in the Declaration of Helsinki and other relevant ethical frameworks.

Minor comments:

Comment 1: Please write the full name for AKT signalling.

Response 1: The full name of AKT – protein kinase B – was added in the first time it appears in the text (introduction section).

Comment 2: For ref 11, could you please cite the main article rather than the preprint?

Response 2: According with the reviewer suggestion, the ref11 was altered to cite the main article and not the preprint.

Comment 3: Please add the experimental details of using ScanProsite tool.

Response 3: Experimental details about ScanProsite tool analysis were added in the “2.1. Bioportides design and synthesis” subsection: “In brief, the CAV1 sequence was retrieved from Uniprot (Uniprot ID: Q03135) and input into ScanProsite. The tool scanned for all types of PP1-binding motifs, identifying the CAV1 PP1-binding motif as an RVxF motif, characterized by the pattern: [RK]-X(0,1)-[VI]-{P}-[FW]”

Comment 4: What is meant by the hydrogen of the main chain nitrogen (HN) and the main chain oxygen (O) of those three residues? Are they the backbone O and N?

Response 4: Yes, the review is correct. Thus, we changed “main chain” to “backbone” in the manuscript.

Comment 5: The caption of Figure 4 is not directly under the figure, please fix.

Response 5: The caption was adjusted to be directly under Figure 4.

Comment 6: In Figure 4 A, the Mg is difficult to be seen, can enrich the sphere’s colour a bit please.

Response 6: The Figure 4 was improved according with the reviewer’s suggestion.

Comment 7: There is no description for the figures in SI.

Response 7: The supplementary figures and tables are stated along the manuscript. A brief description of the figures and tables was added in the captions.

Comment 8: I encourage the author to submit a graphical abstract. In my opinion Figure 8 can be used as GA instead.

Response 8: As encouraged by the reviewer, we submit a graphical abstract.

Reviewer 2 Report

Comments and Suggestions for Authors

The authors have conducted a commendable study on the inhibitory effects of the CAVPENET peptide on prostate cancer cell proliferation and migration through PP1γ-dependent inhibition of AKT signaling. The research is thorough, and the experimental design is robust, demonstrating a clear understanding of the molecular mechanisms involved. The findings significantly contribute to the growing body of knowledge on targeted therapies for prostate cancer and have the potential to inform future therapeutic strategies. Some of the comments that can improve the paper are below:

1.      Authors need to improve the abstract by mentioning the future scope of the paper. They can do so by deleting the line “In conclusion…..”.

2.      Introduction need to improved by mentioning some of the recent FDA approved drugs that target AKT signaling. Also recently various plant metabolites have been studied and characterized for their effectiveness in targeting AKT pathway, these papers can be useful, if yes cite them: https://doi.org/10.3390/biom13020194

 10.2174/0113895575270904231129062137

https://doi.org/10.3390/life13071532

3.      Kindly don’t use the words like ‘We” “I”, check throughout the manuscript.

4.      The methodology should be in sync with the results, maintain the consistency.

5.      Reference style needs to be rechecked and I would encourage authors to add some recent studies particularly from 2024 to make the paper introduction and discussion strong and UpToDate.

6.      Check the figure 2 caption, the alignment is wrong.

7.      Authors should mention which statistical analysis was performed and how many times the experiments were performed to validate the data.

8.      Conclusion needs to be rewritten. The figure citation is conclusion is not needed. Conclusion should be more focused on the research gap filled and future scope of the research.

Comments on the Quality of English Language

Minor corrections needed

Author Response

The authors have conducted a commendable study on the inhibitory effects of the CAVPENET peptide on prostate cancer cell proliferation and migration through PP1γ-dependent inhibition of AKT signaling. The research is thorough, and the experimental design is robust, demonstrating a clear understanding of the molecular mechanisms involved. The findings significantly contribute to the growing body of knowledge on targeted therapies for prostate cancer and have the potential to inform future therapeutic strategies. Some of the comments that can improve the paper are below:

Comment 1: Authors need to improve the abstract by mentioning the future scope of the paper. They can do so by deleting the line “In conclusion…..”.

Response 1: Thank you for the valuable suggestion. The following phrase highlighting the future scope of the work was added in the abstract: “Our results underscore the potential of the designed peptide as a novel therapy for PCa patients, setting the stage for further testing in more advanced models to fully realize its therapeutic promise.”

Comment 2: Introduction need to improved by mentioning some of the recent FDA approved drugs that target AKT signaling. Also recently various plant metabolites have been studied and characterized for their effectiveness in targeting AKT pathway, these papers can be useful, if yes cite them: https://doi.org/10.3390/biom13020194

 10.2174/0113895575270904231129062137

https://doi.org/10.3390/life13071532

Response 2: In response to the reviewer’s suggestion, and to better emphasize the potential of AKT-targeting therapies, we have included additional information in the discussion section “Although the AKT signaling has been thoroughly investigated as a promising target for cancer therapy, with several peptide AKT inhibitors documented in the literature and some of them approved by FDA for cancer treatment68–71, its exploration in PCa remains scarce. Moreover, the AKT-targeting peptides described in the literature directly target AKT or the interaction with co-activators or substrates. Here, we propose an indirect targeting of AKT, by specifically modulating the activity of PP1”

Comment 3: Kindly don’t use the words like ‘We” “I”, check throughout the manuscript.

Response 3: The manuscript was thoroughly reviewed to eliminate the use of the words “We” and “I”.

Comment 4: The methodology should be in sync with the results, maintain the consistency.

Response 4: The manuscript was checked for consistency between methodology and results sections.

Comment 5: Reference style needs to be rechecked and I would encourage authors to add some recent studies particularly from 2024 to make the paper introduction and discussion strong and UpToDate.

Response 5: During the revision process, we added several references to the manuscript including some from the current year.

Comment 6: Check the figure 2 caption, the alignment is wrong.

Response 6: The figure 2 caption was adjusted.

Comment 7: Authors should mention which statistical analysis was performed and how many times the experiments were performed to validate the data.

Response 7: The manuscript includes a “Statistical analysis” subsection within the Methods section, where we detail the statistical tests used throughout the study. Regarding the number of replicates per experiment, we generally indicated in the statistical section that at least three independent replicates were conducted. Nevertheless, and following the reviewer suggestion, we have reviewed all figure captions to ensure the specific number of replicates is provided and have added this information where it was previously missing.

Comment 8: Conclusion needs to be rewritten. The figure citation is conclusion is not needed. Conclusion should be more focused on the research gap filled and future scope of the research.

Response 8: Based on the reviewer’s suggestion, the conclusion was rewritten to better elucidate the benefits of our approach in comparison with others reported in the literature and to emphasize the future scope of the work.

Reviewer 3 Report

Comments and Suggestions for Authors

Review Report

On

CAVPENET peptide inhibits prostate cancer cells proliferation and migration through PP1γ-dependent inhibition of AKT signaling

The work entitled “CAVPENET peptide inhibits prostate cancer cells proliferation and migration through PP1γ-dependent inhibition of AKT signaling” is interesting. It presents a study investigating the anticancer potential of a novel peptide, CAVPENET, which targets protein phosphatase 1 (PP1) complexes to inhibit prostate cancer (PCa) cell proliferation and migration. However, the author needs to make some significant improvements in the manuscript.

Comments

1.     The concept of using peptides to modulate PP1 activity and the targeting of the AKT signaling pathway in cancer therapy is not novel. The manuscript needs to clarify how the CAVPENET peptide provides a significant advancement over existing therapies or research. The authors should provide a more in-depth discussion of how this work differs from and improves upon previous studies.

2.     The manuscript claims that CAVPENET inhibits PCa progression by targeting PP1. However, the specific advantage of this approach over existing therapies is not sufficiently demonstrated. The authors should compare the efficacy of CAVPENET with other known inhibitors of the AKT pathway or other PP1-targeting strategies.

3.     The study relies entirely on in vitro models (PC-3 and LnCaP cell lines), which limits the applicability of the findings. In vivo studies, or at least some preliminary data from animal models, are necessary to validate the therapeutic potential of CAVPENET.

4.     The proposed mechanism by which CAVPENET affects AKT signaling through PP1γ is interesting but not sufficiently supported by the data. The manuscript would benefit from more direct evidence of how CAVPENET interacts with PP1γ, such as co-immunoprecipitation experiments or mutational analysis of the PP1γ binding site.

5.     The study does not sufficiently address the potential off-target effects of CAVPENET, especially given the wide range of processes regulated by PP1. The authors should include experiments or provide a discussion on how they minimized and evaluated off-target effects.

6.     Some of the figures (e.g. Fig. 1) particularly those showing the results of the wound healing and immunoblotting assays, are not of sufficient quality for publication. Improve the resolution of the figures.

7.     While the study focuses on prostate cancer cells, the broader applicability of the findings is not addressed. The manuscript should discuss whether CAVPENET might be effective against other cancer types or if its action is specific to prostate cancer. This would enhance the impact and relevance of the study.

8.     Revise the manuscript for typographical, grammatical, and formatting mistakes.

Comments on the Quality of English Language

Revise the manuscript for typographical, grammatical, and formatting mistakes.

Author Response

“CAVPENET peptide inhibits prostate cancer cells proliferation and migration through PP1γ-dependent inhibition of AKT signaling”

 The work entitled “CAVPENET peptide inhibits prostate cancer cells proliferation and migration through PP1γ-dependent inhibition of AKT signaling” is interesting. It presents a study investigating the anticancer potential of a novel peptide, CAVPENET, which targets protein phosphatase 1 (PP1) complexes to inhibit prostate cancer (PCa) cell proliferation and migration. However, the author needs to make some significant improvements in the manuscript.

Comments

Comment 1: The concept of using peptides to modulate PP1 activity and the targeting of the AKT signaling pathway in cancer therapy is not novel. The manuscript needs to clarify how the CAVPENET peptide provides a significant advancement over existing therapies or research. The authors should provide a more in-depth discussion of how this work differs from and improves upon previous studies. The manuscript claims that CAVPENET inhibits PCa progression by targeting PP1. However, the specific advantage of this approach over existing therapies is not sufficiently demonstrated. The authors should compare the efficacy of CAVPENET with other known inhibitors of the AKT pathway or other PP1-targeting strategies.

Response 1: Following the reviewer suggestion, the following information highlighting the novelty of CAVPENET, comparing with other AKT-targeting peptides was added to the Discussion section: “Although the AKT signaling has been thoroughly investigated as a promising target for cancer therapy, with several peptide AKT inhibitors documented in the literature69–71, its exploration in PCa remains scarce. Moreover, the AKT-targeting peptides described in the literature directly target AKT or the interaction with co-activators or substrates. Here, we propose an indirect targeting of AKT, by specifically modulating the activity of PP1”.

These ideas were also highlighted in the conclusion section by adding the following text: “This approach allows for a more selective modulation of AKT signaling, compared with others reported in the literature. In addition to minimize toxic effects, potentially enables a tissue-specific effect.”

Comment 2: The study relies entirely on in vitro models (PC-3 and LnCaP cell lines), which limits the applicability of the findings. In vivo studies, or at least some preliminary data from animal models, are necessary to validate the therapeutic potential of CAVPENET.

Response 2: We have some preliminary data on the effect of CAVPENET peptide in a more advanced 3D in vitro model, which will be published in a separated manuscript, when concluded.

In that work, we assess the effect of CAVPENET on co-culture 3D spheroids, composed of prostate cancer cells and prostate cancer-associated fibroblasts, and we observe a significant decrease in spheroids viability, while no significant effect was noted in monoculture spheroids only composed by CAFs, highlighting the therapeutic potential of the peptide.

Comment 3: The proposed mechanism by which CAVPENET affects AKT signaling through PP1γ is interesting but not sufficiently supported by the data. The manuscript would benefit from more direct evidence of how CAVPENET interacts with PP1γ, such as co-immunoprecipitation experiments or mutational analysis of the PP1γ binding site.

Response 3: Thank you for your insightful comment. Incorporating co-immunoprecipitation or mutational analysis of PP1γ binding site would indeed add significant value to our research. We will certainly consider integrating these experimental approaches in future studies, potentially within the context of our 3D co-culture model.

Comment 4: The study does not sufficiently address the potential off-target effects of CAVPENET, especially given the wide range of processes regulated by PP1. The authors should include experiments or provide a discussion on how they minimized and evaluated off-target effects.

Response 4: Thank you for your comment. Indeed, we did not focus on off-target effects of CAVPENET peptide. As the reviewer stated, the PP1 is known to be involved in a wide range of cellular processes and targeting PP1 was associated with toxicity in a variety of studies reported in the literature. In this context, emerged the idea of target specific PP1 complexes as a more specific and with minimized toxic effects. In this work, we used this approach, aiming to target a PP1 complex (PP1/CAV1), that was only previously identified in prostate and colorectal cancer cells. To improve the specificity of CAVPENET peptide, we can consider adding a homing peptide to the sequence, thus minimizing the off-target effects.

Added in the manuscript in the Discussion section “Targeting PP1 active site was firstly proposed as a promising therapeutic approach. The potential therapeutic effect of PP1 inhibitors tautomycin and calyculin A was evaluated in PCa cells. Despite effective in causing cancer cells apoptosis, both were associated with significant toxic effect, limiting their applicability55,56. More recently, emerged the concept of target specific PP1 complexes as a more selective approach, allowing the modulation of PP1 activity against specific substrates”.

Comment 5: Some of the figures (e.g. Fig. 1) particularly those showing the results of the wound healing and immunoblotting assays, are not of sufficient quality for publication. Improve the resolution of the figures.

Response 5: According to the reviewer’s suggestion, the resolution of the figures was improved.

Comment 6: While the study focuses on prostate cancer cells, the broader applicability of the findings is not addressed. The manuscript should discuss whether CAVPENET might be effective against other cancer types or if its action is specific to prostate cancer. This would enhance the impact and relevance of the study.

Response 6: PP1 complexes highly depend on the tissue and biological context. To the best of our knowledge, the PP1/CAV1 was only identified in prostate and colorectal cancer. Thus, we expect that CAVPENET may have a direct impact in colorectal cancer. Moreover, this approach may be easily adapted for other PP1 complexes, thus allowing the application in other types of cancer.

Added in the discussion section of the manuscript: “In addition to its demonstrated potential in treating PCa, CAVPENET may also be applicable to other cancer types. For instance, the target complex PP1/CAV1 has already been identified in colorectal cancer cells104, suggesting that CAVPENET could be directly applicable for colorectal cancer treatment. Moreover, the approach outlined in this study could be adapted to target other PP1 complexes that play critical roles in the progression of various cancers.”

Comment 7: Revise the manuscript for typographical, grammatical, and formatting mistakes.

Response 7: The manuscript was revised for typographical, grammatical and formatting issues.

Round 2

Reviewer 1 Report

Comments and Suggestions for Authors

Dear Authors,

Congratulations for the great work, really!

In fact my fifth comment was whether you checked the peptides (without) the CPP vehicle (penetratin)? but mistakenly, I typed with.

The caption of Figure 4 still not under the figure, maybe the layout, fine the journal could fix this.

The Mg Sphere still not clear, maybe this the best that could be exported from the software.

No problem at all, those comments just to enhance the final version only, I recommend publishing your cool research!

Regards

Reviewer 2 Report

Comments and Suggestions for Authors

Authors have made several changes to the manuscript which has improved the quality of the manuscript, thus can be considered for publication.

Reviewer 3 Report

Comments and Suggestions for Authors

No More comments, authors adequately revised the manuscript